# The Effects of an Osteoarthritic Joint Environment on ACL Damage and Degeneration: A Yucatan Miniature Pig Model

**DOI:** 10.3390/biom13091416

**Published:** 2023-09-20

**Authors:** Elias Schwartz, Kenny Chang, Changqi Sun, Fei Zhang, Guoxuan Peng, Brett Owens, Lei Wei

**Affiliations:** Department of Orthopaedics, Warren Alpert Medical School of Brown University, Rhode Island Hospital, Providence, RI 02903, USA; elias_schwartz@brown.edu (E.S.); kenny_chang@brown.edu (K.C.); changqi_sun@brown.edu (C.S.); fei_zhang@brown.edu (F.Z.); pgxuan@outlook.com (G.P.); brett_owens@brown.edu (B.O.)

**Keywords:** posttraumatic osteoarthritis, inflammation, minipig, anterior cruciate ligament

## Abstract

Posttraumatic osteoarthritis (PTOA) arises secondary to joint injuries and is characteristically driven by inflammatory mediators. PTOA is often studied in the setting of ACL tears. However, a wide range of other injuries also lead to PTOA pathogenesis. The purpose of this study was to characterize the morphological changes in the uninjured ACL in a PTOA inflammatory environment. We retrospectively reviewed 14 ACLs from 13 Yucatan minipigs, 7 of which had undergone our modified intra-articular drilling (mIAD) procedure, which induced PTOA through inflammatory mediators. Seven ACLs were harvested from mIAD minipigs (PTOA) and seven ACLs from control minipigs with no cartilage degeneration (non-PTOA). ACL degeneration was evaluated using histological scoring systems. IL-1β, NF-κB, and TNF-α mRNA expression in the synovium was measured using qRT-PCR. PTOA minipigs demonstrated significant ACL degeneration, marked by a disorganized extracellular matrix, increased vascularity, and changes in cellular shape, density, and alignment. Furthermore, IL-1β, NF-κB, and TNF-α expression was elevated in the synovium of PTOA minipigs. These findings demonstrate the potential for ACL degeneration in a PTOA environment and emphasize the need for anti-inflammatory disease-modifying therapies following joint injury.

## 1. Introduction

Posttraumatic osteoarthritis (PTOA) comprises approximately 12% of all osteoarthritis cases and arises secondary to joint trauma or injury [1]. In particular, a slew of common orthopedic conditions, such as anterior cruciate ligament (ACL) tears, meniscal injuries, and patellar dislocations, have been identified as risk factors potentiating the development of PTOA [2]. Following an injury to the joint, a complex interaction of mechanical and biological factors leads to the development of cartilage degeneration. Unfortunately, surgical treatment to restore the mechanics of the joint cannot fully reduce the risk of PTOA development. For instance, 73% of individuals suffering an ACL injury who subsequently undergo surgical repair still develop PTOA within 10 to 15 years after their primary injury [3]. Research has established that the characteristic inflammatory profile of PTOA—elevated levels of inflammatory mediators, such as IL-1β, NF-κB, and TNF-α, and catabolic enzymes, such as matrix metalloproteinases (MMPs)—drives the pathogenic changes observed in the injured joint [4]. 

There is a growing focus on isolating and better understanding the contribution of inflammation in the absence of demonstrable mechanical dysfunction following joint injury [4]. Huebner et al. demonstrated that drilling osseous tunnels in non-articulating and non-load-bearing locations could consistently induce PTOA in rabbits while sparing the ligamentous structures [5]. Despite the absence of intraoperative damage to soft tissues, the evaluation of cartilage integrity at 3, 6, 9, and 52 weeks following bone tunneling revealed dramatically increased cartilage degeneration, synovial infiltration, and pro-inflammatory cytokines [5]. In addition, Sun et al. recently demonstrated that inflammation drives the development of PTOA following the osseous drilling of the tibia and femur adjacent to intact, unmanipulated ACLs in the Yucatan minipig [6]. Despite the absence of intraoperative damage to soft tissues and no detectable gait disturbance, the evaluation of cartilage integrity at 15 weeks following bone tunneling revealed dramatically increased cartilage degeneration and synovial inflammatory cells and cytokines compared with a sham surgery group. Although the impact of inflammation has been extensively documented to contribute to the progressive decay of articular cartilage, there is a limited amount of literature regarding the effects of inflammation on the neighboring soft tissue, such as the ACL. Previously, Otani et al. found that elderly patients undergoing total knee arthroplasty for long-term osteoarthritis had degenerative changes in their meniscus that were associated with inflammatory cytokine (IL-6 and IL-8) levels [7]. It is important to understand these changes because damage to the soft tissue architecture secondary to posttraumatic inflammation may subtly impair movement mechanics, place the patient at greater risk of additional ligamentous injury, and portend further PTOA progression [8,9]. Previously, animal models induced PTOA by creating permanent deficiencies of the ACL or meniscus, which limits the ability to control for mechanical changes and study the isolated effects of inflammation on these structures. 

As a result, the purpose of this study was to perform a modified intra-articular drilling (mIAD) procedure in the Yucatan minipig, previously developed by Sun et al., to induce inflammation characteristic of PTOA and evaluate changes to previously healthy, undamaged ligaments. We hypothesized that, even in the absence of any direct acute surgical trauma to the ACL, intra-articular ACLs of the bone-drilled knees would exhibit chronic degenerative changes, marked by extracellular matrix organization, vascularity of the ligament, glycosaminoglycan distribution and density, and cellular shape, distribution, and alignment, due to exposure to increased local mediators of inflammation in the PTOA environment. 

## 2. Materials and Methods

### 2.1. Study Design

This study was approved by the Institutional Animal Care and Use Committees of our institution and the Animal Care and Use Review Office from the Department of Defense (Protocol number: 19-04-0001). This study was designed to meet the ARRIVE guidelines [10]. We retrospectively reviewed 14 ACLs that were harvested from 13 minipigs; 7 minipigs underwent our previously developed modified intra-articular drilling (mIAD procedure), and 7 ACLs were taken unilaterally from the surgical knees from our previous study [6]. Six minipigs underwent sham surgery, five ACLs were taken unilaterally, and two ACLs were taken bilaterally. All animals assigned to the non-PTOA group were confirmed with no macroscopic cartilage changes, and all animals in the PTOA group received the mIAD procedure and showed significant cartilage degeneration, as confirmed by macroscopic changes. All animals were housed individually in adjacent pens (minimum pen size of 2.1 m^2^) for 15 weeks. It has been demonstrated that 15 weeks is sufficient time to observe both macroscopic and microscopic cartilage changes indicative of PTOA [6,11]. 

### 2.2. Surgical Technique

The animals were anesthetized and a medial arthrotomy was performed on the left hind knee to access the ACL. A commercial drill system (Cordless Driver 4; Stryker, Kalamazoo, MI, USA) was used to perform the drilling procedure with Kirschner wires. Two osseous tunnels that were 15 mm deep with 2 mm diameter were drilled into the tibial bone adjacent to the posterior and anterior edges of the tibial ACL insertion (Figure 1). The drilling procedure was repeated at the posterior lateral and anterior medial edges of the femoral ACL insertion similarly to the procedure previously described by Huebner et al. in rabbits [5]. The wound was then irrigated with sterile saline. The incision site was closed in layers with buried absorbable sutures (Vicryl, Ethicon, Raritan, NJ, USA): Arthrotomy: 0, interrupted; Bursa: 2-0, running; subcutaneous tissue: 2-0, interrupted; and subcuticular layer: 3-0 interrupted. A fentanyl patch was placed on the dorsum of the pigs for 3 days post-surgery to manage post-operative pain. The animals were treated with Ondansetron, an anti-vomiting agent, immediately post-operation. Additionally, the animals were kept in pens for the duration of the study. They were not forced to exercise, though appeared active when checked daily. The sham control group underwent the same arthrotomy procedure and closure, but without the intra-articular drilling. The animals were euthanized with a euthanasia solution (Beuthanasia-D Special, Merck, Madison, WI, USA) at 15 weeks, and both hind knees were harvested. 

### 2.3. Macroscopic Cartilage Assessment

The lateral and medial articulating surfaces of the femoral condyle and tibial plateau of all the studied limbs were stained with India ink and photographed. The length and width of India ink-stained lesions were measured using calipers and the lesion area was approximated as an ellipse [12]. Based on the macroscopic cartilage changes, ACL samples were categorized into non-PTOA and PTOA groups. If observable macroscopic cartilage damage was present, ACLs were categorized into the PTOA group, and ACLs from pigs with no macroscopic cartilage damage were categorized into the non-PTOA group. ACLs from the PTOA group included all mIAD surgical legs (*n* = 7). 

### 2.4. Cartilage Histological Preparation

Osteochondral samples were extracted from the central medial and lateral compartments of both the tibial plateau and femoral condyles within the knee joint. These samples encompassed the complete width of the tibial plateau and femoral condyles, with distinct separation of the medial and lateral aspects. Following extraction, the samples underwent fixation in a 10% formalin solution for a duration of 48 h, and were subsequently preserved in a 70% ethanol solution. The samples were subjected to a 30-min irrigation under tap water. Post-cleansing, the specimens were placed within cassettes and exposed to a buffered formic acid decalcification solution (20% formic acid and 10% sodium citrate in dH2O) at room temperature. To enhance reagent penetration, any bone material outside the region of interest was incised using a razor blade. On the third day, the samples were bifurcated longitudinally. The decalcification solution was changed every 3–4 days, with the progress of decalcification assessed using X-ray imaging. The decalcification process was considered complete within a span of 7–10 days, contingent upon the dimensions of the tissue. After decalcification, the specimens were rinsed in running tap water for another 30 min, then placed in a 70% ethanol solution at room temperature. The specimens then underwent a dehydration and paraffin infiltration process using a pressure/vacuum-driven automated tissue processor. Following successful infiltration, the tissue was tailored to fit embedding molds, ensuring that the bifurcated portions were oriented downward within the mold. The embedded blocks were safeguarded from light and stored in filing boxes at room temperature. For sectioning, the embedded samples were positioned on a rotary microtome and sliced into 6 μm thick sections, which were subsequently mounted onto positively charged slides.

### 2.5. Microscopic Cartilage Assessment

To confirm the osteochondral degenerative changes observed macroscopically, cartilage tissue was stained with Safranin O-fast green and was scored according to a modified Osteoarthritis Research Society International (OARSI) grading system for characterizing the severity of cartilage degeneration in large animals [13]. Four parameters were scored using 6 blinded evaluators: structure (0–10), chondrocyte density (0–4), cell cloning (0–4), and interterritorial Safranin O-fast green (0–4). A higher number indicates more severe cartilage degeneration. 

### 2.6. ACL Histological Preparation

Segments of the ACL were dissected from each knee, and 2 ACL tissue sections were cut per knee using a microtome. A total of 28 tissue sections were prepared from 14 harvested ACLs. The samples were then fixed in 10% formalin for 48 h and stored in 70% ethanol. The samples were then rinsed in running water and transferred to 70% ethanol at RT. Tissue was loaded onto an automated tissue processor for dehydration and paraffin infiltration. After infiltration, the tissue was trimmed to fit into embedding molds, and embedded blocks were stored in a filing box protected from light at RT. The embedded samples were sectioned with a rotary microtome at 6 μm onto slides. The tissue was minimally trimmed before collection (around 100 µm). Before staining, the slides underwent deparaffinization via heating for two hours to melt the paraffin, followed by an ethanol gradient. The slides were stained with hematoxylin and eosin (H&E) and Alcian blue. 

### 2.7. Microscopic ACL Assessment

Slides stained with H&E and Alcian blue were scored according to a modified grading system from Kharaz et al. [14]. For H&E staining, the grading items included the extracellular matrix organization of the whole ligament (0–2), cellular shape (0–2), cellular distribution (0–1), cellular alignment (0–2), and vascularity of the ligament (0–1). For Alcian blue staining, the graded items included the glycosaminoglycan (GAG) distribution and density (0–3), cellular shape (0–2), cellular distribution (0–1), and cellular alignment (0–2). The samples were scored by 3 blinded evaluators. Lower scores indicated greater degenerative changes. Additional microscopic quantitative analysis was performed through cell counting. The count feature was utilized on a microscope (Nikon Ni-E, NIS-Elements AR. DS-U2/L2-Ril software, Brighton, MI, USA) to count all cell nuclei in a constant frame (418.34 μm × 334.29 μm). Three distinct frames were captured randomly for each slide, and the average cell density was calculated for each measurement.

### 2.8. Quantitative Real-Time PCR for Inflammatory Mediators

The gene expression levels of IL-1β, NF-κB, and TNF-α from synovial membranes were analyzed using the quantitative real-time reverse transcriptase–polymerase chain reaction (qRT-PCR) according to published protocols (iQ SYBR Green Supermix, Bio-Rad, Hercules, CA, USA). Synovial membrane tissues were homogenized in TRIzol reagent (cat# 15596026, Invitrogen, Waltham, MA, USA) using a homogenizer. Subsequently, the total RNA was extracted with the TRIzol reagent and further purified using an RNeasy Mini Kit (cat# 74004, Qiagen, Hilden, Germany). Gene expression was measured through two methods: first-strand cDNA synthesis using a reverse transcription kit (cat# 1708890, Bio-Rad, Hercules, CA, USA) and qRT-PCR using the iQTM SYBR Green Supermix kit (cat# 170-8887, Bio-Rad, Hercules, CA, USA) in a real-time PCR system (CFX Connect, Bio-Rad, Hercules, CA, USA). A total of 1 μg of RNA was used for cDNA synthesis. 

Priming was conducted at 25 °C for 5 min. Reverse transcription was conducted at 40 °C for 20 min. Reverse transcriptase inactivation was conducted at 95 °C for 1 min. For qRT-PCR testing, 18 s rRNA, an established reference gene, was used as an internal control [15]. Swine-specific primers were designed and synthesized by Integrated DNA Technologies (IDT, Coralville, IA, USA), as specified in Table 1. PCR was performed for 40 cycles after an initial denaturation step at 95 °C for 3 min. Each cycle involved an additional denaturation step for 15 s at 95 °C, annealing for 60 s at 55 °C or 58 °C, and extension for 40 s at 72 °C. The reaction was terminated at 70 °C after a 10-min extension. Three independent PCR experiments were performed to obtain the relative level of expression for each gene (IL-1β, NF-κB, TNF-α).

### 2.9. Immunohistochemistry

The ACL specimens were analyzed using the immunohistochemistry Detective System (TL-015-HD, Epredia, Kalamazoo, MI, USA) to assess the IL-1β and NF-κB levels. Briefly, sections were digested with 5 mg/mL of hyaluronidase in phosphate-buffered saline. Sections were then incubated with 1:100 of antibody against rabbit IL-1β (cat#420B, Invitrogen, Waltham, MA, USA) and 1:100 of antibody against rabbit NF-κB (cat#10745-1-AP Proteintech Group, Rosemont, IL, USA) at 4 °C overnight. Coverslips were mounted and visualized under the microscope. 

### 2.10. Statistical Methods

Data were imported into statistical software (SAS version 9.4; SAS Institute Inc., Cary, NC, USA) for hypothesis testing. Student’s *t*-tests were used to compare differences in the macroscopic damage, ACL hematoxylin and eosin and Alcian blue histological scores, and qRT-PCR levels between the PTOA and non-PTOA groups. An adjusted *p*-value of <0.05 was used to determine statistical significance. 

## 3. Results

### 3.1. Macroscopic Cartilage Assessment

Macroscopic cartilage degeneration, as an indicator of PTOA, was measured using India ink staining. For the minipigs selected for the non-PTOA group, there was no observable damage on either the femoral condyles or tibial plateaus (0 ± 0 mm^2^) (Figure 2A). The minipigs selected for the PTOA group exhibited observable changes in cartilage integrity, with visible lesions on both the femoral condyles and tibial plateaus (36.4 ± 21.5 mm^2^) (Figure 2A). There was a statistically significant difference in lesion size between non-PTOA and PTOA groups (*p* < 0.001) (Figure 2B). Lesions were predominantly located in the medial femoral condyle (MFC) (14.0 ± 9.84 mm^2)^ and medial tibial plateau (MTP) (17.8 ± 11.5 mm^2^), with minimal lesions present in the lateral femoral condyle (LFC) (4.49 ± 7.88 mm^2^) and lateral tibial plateau (LTP) (0.112 ± 0.297 mm^2^).

### 3.2. Microscopic Cartilage Assessment

In the medial compartment of the PTOA knee, we observed marked degenerative changes in the cartilage characterized by decreased chondrocyte density, increased chondrocyte clusters, surface irregularities, erosions to the deep zone, and decreased interterritorial Safranin O staining to the middle zone (Figure 3A). In comparison, non-PTOA knees exhibited minimal changes in chondrocyte density, minimal chondrocyte clusters, few surface irregularities, and no changes in interterritorial Safranin O staining or erosions to the deep zone (Figure 3B). The total severity of the cartilage damage score was significantly higher in the PTOA group compared with the non-PTOA group (*p* < 0.0001) (Figure 3C).

### 3.3. Microscopic ACL Assessment

The severity of microscopic ACL damage was significantly greater in the PTOA joint than that in the non-PTOA joint (*p* < 0.05) (Figure 4A–D, Figure 5A–F, and Figure 6A–D). Lower scores indicated greater degeneration. In general, non-PTOA joints showed healthy ACL tissue interfascicular matrix (IFM) and extracellular matrix organization (ECM), indicated by long, compact collagen fibers (Figure 4A). ACLs in the PTOA group exhibited significantly greater disorganization, with loosely composed and unordered collagen fibers (*p* = 0.001) (Figure 4B and Figure 5A). In the non-PTOA group, although most cells displayed a healthy spindle shape, there was a heterogeneous mix of both spindle and round cells (Figure 4B). In the PTOA group, cells exhibited a loss of the spindle shape and were significantly more elliptical and circular in shape (*p* = 0.002) (Figure 4B and Figure 5B). Cell alignment was measured via orientation and distribution along collagen fibers, and there was an observable difference along the IFM when comparing non-PTOA and PTOA joints (Figure 4A–C). In the PTOA group, cells exhibited a loss of uniaxial alignment and were significantly less uniform in orientation and distribution (*p* = 0.001) (Figure 5C). Cell distribution was measured based on focal areas of elevated cell density (cell clustering or formation of cell chains (Figure 4B,C). In the PTOA joint, cells exhibited significant aggregation and formation of cell chains (*p* < 0.001) (Figure 5D). Vascularization was measured as either hypo-vascularized or hyper-vascularized (increased number of capillaries with cellular infiltrate). ACLs in the PTOA group exhibited significant hyper-vascularization with cellular infiltrate (*p* = 0.005) (Figure 4C and Figure 5E). Overall, aggregating the scores of each parameter yielded a statistically significant difference between non-PTOA and PTOA joints (*p* = 0.001) (Figure 5F). The cell density of the ACL in the PTOA joint was significantly greater than that in the non-PTOA joint. (*p* < 0.001) (Figure 4D). 

Alcian Blue staining and a modified scoring system based on Kharaz et al. were used to quantify the differences in glycosaminoglycan (GAG) accumulation between non-PTOA and PTOA joints [14]. ACLs in the PTOA group exhibited significantly increased GAG accumulation, with darker blue staining around rounded cells as an indication (Figure 6A,B). GAG distribution and density were measured as percentages, and there was a significant difference found between groups based on the score (*p* = 0.012) (Figure 6D). Overall, in aggregates with the same parameters used for the H&E staining (cell shape, alignment, and distribution), there was a significant difference between non-PTOA and PTOA joints (*p* < 0.001) (Figure 6C). 

### 3.4. Inflammatory Mediators

The RT-PCR results indicated changes in the levels of inflammatory mediators (IL-1β, NF-κB, and TNF-α mRNA expression levels) present between non-PTOA and PTOA joints (Figure 7). The IL-1β levels in PTOA knees were, on average, four times those in non-PTOA knees (*p* = 0.04). The NF-κB levels in PTOA knees were, on average, five times those in non-PTOA knees (*p* = 0.004). The TNF-α levels in PTOA knees were, on average, seven times those in non-PTOA knees (*p* < 0.001). 

### 3.5. Immunohistochemistry

Immunohistochemistry staining for NF-κB and IL-1β indicated increased inflammation between PTOA and non-PTOA groups. Non-PTOA groups demonstrated no positive staining for either NF-κB or IL-1β (Figure 8). Staining density was the highest surrounding the blood vessels and cell aggregates present in PTOA groups. 

## 4. Discussion

Patients who sustain an intra-articular joint injury are at greater risk of suffering from future cartilage degeneration due to a complex interaction between mechanical and biological factors [2]. Beyond ACL tears, lesions to the meniscus and posterior cruciate ligament, as well as intra-articular osseous fractures, are common injuries requiring surgical treatment [2,16,17,18,19]. Uncontrolled inflammation, through the upregulation of TNF-α, IL-1β, and IL-6, remains even after surgical stabilization and places the injured joint in a catabolic environment [20,21]. In minipigs undergoing ACL transection, Siker et al. found decreased GAG density and increased immune cell density in cartilage at 1 and 4 weeks, which was associated with upregulated cytokine expression, including IL-1β and TNF-α [22]. Although the impact of this on articular cartilage is well-documented, the effects of lingering inflammation on the neighboring soft tissue, such as the anterior cruciate ligament (ACL), are not. In commonly used animal models of posttraumatic osteoarthritis (PTOA), such as ACL transection, surgical damage to the ACL is the prime driver of subsequent degenerative changes to both the articular cartilage and ACL [23]. However, the initial and lingering mechanical damage to the ACL does not capture the biomechanics of a surgically restabilized joint and likely overshadows the degenerative effects of inflammatory pathways [24]. 

In this study, we demonstrated that inflammation in the mechanically stable PTOA joint caused significant microscopic ACL degeneration in the Yucatan minipig. In particular, the ACLs of minipigs undergoing mIAD exhibited loosely composed, unorderly extracellular matrix organization, abnormal cellular appearance, hyper-vascularity, increased GAG accumulation, and increased cell density. These changes are indicators of ACL degeneration, as they represent changes toward a stiffer chondrocyte-like collagen matrix and the recruitment of immune cells, such as lymphocytes, neutrophils, and macrophages [25]. ACL degeneration has also been associated with the recruitment and proliferation of myofibroblasts and progenitor cells, likely contributing to the increased cell density we observed in minipigs undergoing mIAD [26]. Additionally, chondroid metaplasia has been associated with ECM disorganization, which can lead to ligament biomechanical failure [8,9]. Furthermore, the synovial membrane of minipigs undergoing mIAD had significantly elevated levels of inflammatory mediators (IL-1β, NF-κB, and TNF-α). By utilizing the mIAD surgical technique, the peri-operative biomechanics of the ACL and other important static stabilizers were preserved, and any subsequent damage was likely caused by biological changes. These findings support our hypothesis that an inflammatory intra-articular environment can lead to significant ACL damage and degeneration within just 15 weeks of injury. Translating to clinical practice, the findings of this study suggest that inflammation from a sufficiently severe intra-articular joint injury can induce additional ligamentous degeneration, even after the surgical stabilization of the initial injury.

Inflammatory mediators are heavily involved in PTOA degenerative processes, with the upregulation of IL-1β, NF-κB, and TNF-α in the synovial membrane playing a key role [27]. Pro-inflammatory cytokines (IL-1β, NF-κB, and TNF-α) have been found in the synovial fluid of patients sustaining ACL injury, as well as patients with established OA [21,28]. Synovial inflammation and the release of pro-inflammatory cytokines stimulate osteoarthritic progression by increasing the expression of matrix metalloproteinases (MMPs), which have been demonstrated to cause fibrillation and the loss of collagen in intra-articular cartilage [29]. Downstream, MMPs target a diverse range of protein substrates and degrade multiple types of collagens, including type I collagen, the major structurally stabilizing form present in the ACL [30]. Additionally, the secretion of IL-1β and TNF-α from the synovial membrane also decreases the synthesis of type II collagen, a key component in maintaining the expression of pro-chondrogenic cytokines [31]. Reflective of the inflammatory profile of PTOA, in our study, the IL-1β, NF-κB, and TNF-α mRNA expression levels were all significantly elevated in the synovial membrane of minipigs in the PTOA group compared with the non-PTOA group. Additionally, immunohistochemistry revealed marked differences between non-PTOA and PTOA groups, with elevated levels of both IL-1βNF-κB in the ACL tissue of PTOA pigs. Consequently, the inflammatory joint environment induced by elevated cytokine levels may stimulate the catabolism of the normal type I collagen in the ACL, leading to the morphological degenerative changes observed in the ACL.

Previous animal and human studies have reported similar histological changes in the ACL, albeit in a mechanically unstable joint environment [32,33]. In a murine model of PTOA, Ramos-Mucci et al. found that the destabilization of the medial meniscus (DMM) induced notable cell hypertrophy, proteoglycan accumulation, and ECM disorganization of the surrounding ligaments, particularly the ACL [33]. The degeneration of the ACL in the DMM animal model is likely a function of both biological and mechanical factors. However, owing to the interdependence between the ACL and the meniscus, a medial meniscal-deficient knee may overburden the ACL with additional strain during the stabilization of anterior tibial translation [34,35]. Given that excess mechanical stress is a cause of morphological changes, it is not possible to isolate the degenerative effects of inflammation in animal models using DMM [36]. Furthermore, in elderly patients with late-stage primary OA and no prior history of ACL injury or surgery, Hasegawa et al. found that the ACL exhibited a decreased alignment of fiber bundles and marked cellular changes, including increased cellular density, aggregates, and a rounded shape [26]. Furthermore, immunohistochemistry showed increased expression of Sox9, a transcription factor involved in potentiating a chondrocyte-like phenotype in degenerated ACLs [26,37]. Our study corroborates these findings, with GAG accumulation in the ACLs of PTOA pigs indicating the presence of chondrocyte-like cell aggregates. Although we observed significant changes in the ACL in our mIAD model, which preserved the meniscus, ACL, and all ligaments in the intra-articular space, these observations are likely attributed to the elevation in inflammatory cytokines from the synovium, rather than mechanical changes that increase stress on the ACL. 

There are several limitations of this study to consider. Macroscopic indications of PTOA damage were measured at 15 weeks post-drilling. Consequently, although there were generally no differences in the mechanical function of the ACL, as measured through gait analysis, when measured at 15 weeks, progressive degeneration of both articular cartilage and ACL integrity may lead to noticeable changes in joint loading at later time points [6]. However, these later changes would be secondary to the inflammatory cascades in an initially mechanically stable joint. An additional limitation of our study was the relatively small sample size of ACLs harvested (*n* = 14) compared with similar studies with humans and animals [26,33,38]. Still, other studies have examined ACL morphology in an osteoarthritic joint environment in other animal models and with mini-pigs with smaller sample sizes [14,39]. 

Future directions involve examining anti-inflammatory therapeutic interventions to prevent damage to associated tissues during the early stage of traumatic joint injury. Traditional anti-inflammatory research for PTOA centers around the effects of corticosteroid and hyaluronic acid intra-articular injections, which have shown a reduction in pain and disease progression in clinical trials [40,41]. However, the majority of studies that have examined the use of both these therapeutic interventions include elderly patient populations with progressive OA, in contrast with younger patients with acute PTOA [42]. Recently, alpha-2-macroglobulin (A2M) has emerged as a potential intra-articular injection therapy for treating PTOA, with in vitro and in vivo studies exhibiting promising anti-inflammatory effects that preserve cartilage integrity [43,44,45]. The chondroprotective effects of A2M therapy may translate to ACL preservation, which we aim to examine in future studies.

## 5. Conclusions

In conclusion, the mIAD procedure, which induces inflammation leading to PTOA in a mechanically stable joint, also leads to the microscopic degeneration of the ACL. These results indicate that the ACL is susceptible to biological factors, which may arise following injuries to other structures of the joint. Further studies should evaluate how anti-inflammatory agents may attenuate soft tissue changes in an inflammatory joint environment characteristic of PTOA.

## Figures and Tables

**Figure 1 biomolecules-13-01416-f001:**
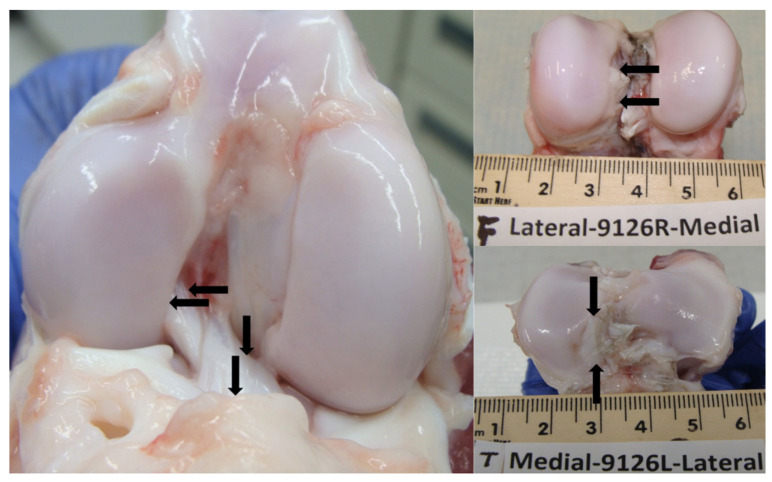
Drill site locations on the femur and tibia are indicated by the black arrows. The femur was drilled 15 mm deep with a 2 mm diameter on the posterior lateral and anterior medial edges of the femoral ACL insertion. The tibia was drilled in the same fashion.

**Figure 2 biomolecules-13-01416-f002:**
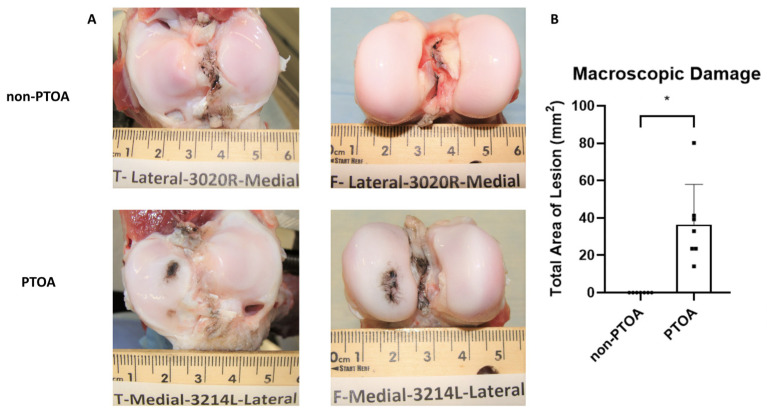
Macroscopic cartilage changes. (**A**) Images of the median lesions area of the tibial plateau and femoral condyle for non-PTOA and PTOA animals. (**B**) Total lesion area for non-PTOA and PTOA groups. Error bars represent standard deviation. * Indicates a significant difference between groups (*p* < 0.05).

**Figure 3 biomolecules-13-01416-f003:**
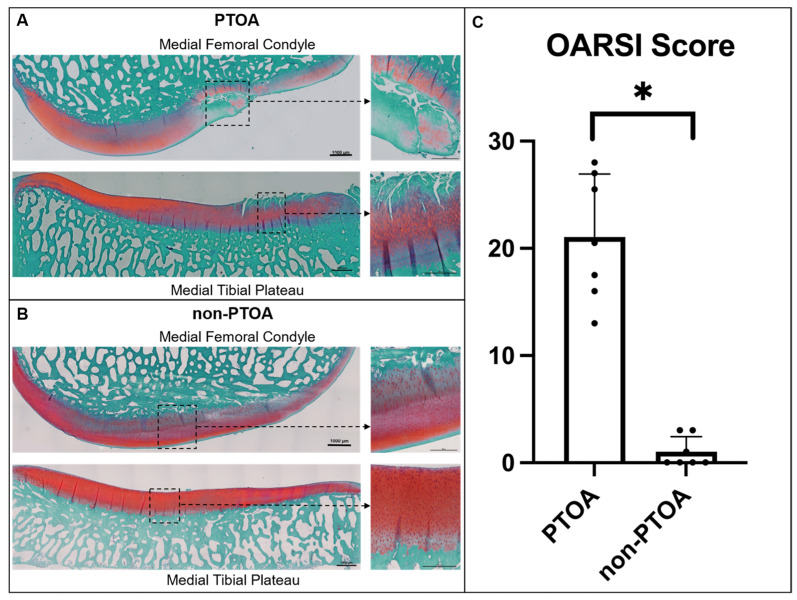
Microscopic cartilage assessment. Histological images of PTOA group’s medial femoral condyle and medial tibial plateau stained with Safranin O-fast green for (**A**) PTOA and (**B**) non-PTOA groups. Representative scale bars of 1000 μm and 500 μm, respectively. (**C**) Severity of cartilage damage indicated by modified OARSI scoring system. Error bars represent the standard deviation. * Indicates a statistically significant difference.

**Figure 4 biomolecules-13-01416-f004:**
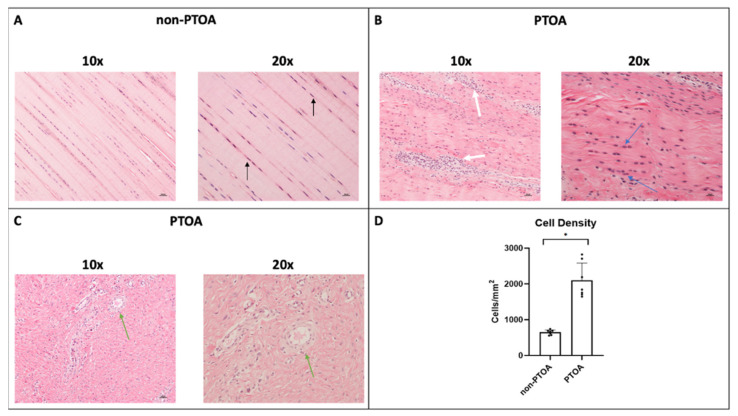
Microscopic changes in ACL. (**A**) Histological staining (H&E) images of the ACL of non-PTOA minipigs at 10× and 20× magnification. Black arrows indicate healthy spindle-shaped cells with uniaxial alignment. (**B**,**C**) Histological images of the ACL of PTOA animals at 10× and 20× magnification. White arrows indicate cell aggregates and lack of alignment. Blue arrows indicate rounded cells. Green arrows indicate hypervascularization with increased cellular infiltrate. (**D**) ACL cell density. Error bars represent standard deviation. * Indicates significant difference between groups (*p* < 0.05).

**Figure 5 biomolecules-13-01416-f005:**
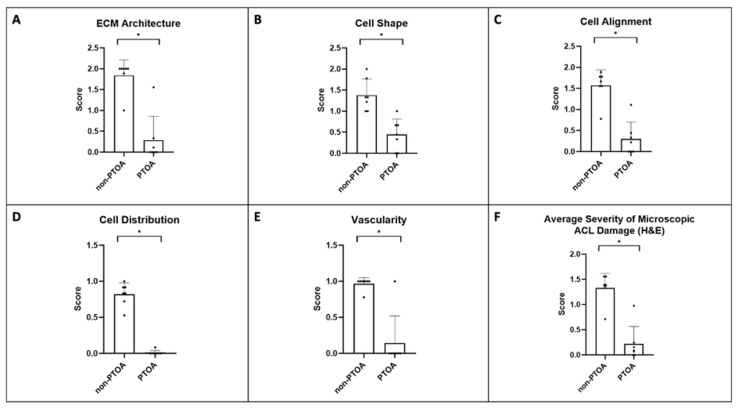
Bar charts of the H&E histological scoring components for the ACL. (**A**) ECM architecture, (**B**) cell shape, (**C**) cell alignment, (**D**) cell distribution, (**E**) vascularity, and (**F**) average severity of microscopic ACL damage. * Indicates significant difference between groups (*p* < 0.05).

**Figure 6 biomolecules-13-01416-f006:**
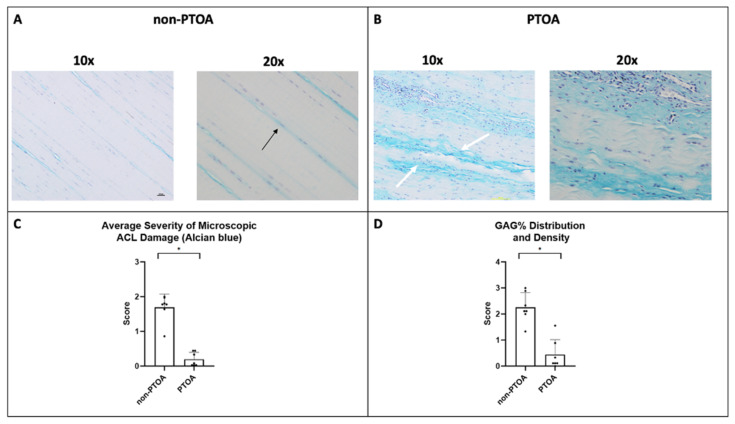
Microscopic changes in ACL. (**A**) Histological staining (Alcian Blue) images of non-PTOA animals at 10× and 20× magnification, observable uniaxial fiber alignment; black arrows indicate spindle-shaped cells. (**B**) Images of PTOA animals at 10× and 20× magnification, white arrows indicate areas of high GAG accumulation. (**C**,**D**) Scores for extent of microscopic damage. * Indicates significant differences between groups (*p* < 0.05).

**Figure 7 biomolecules-13-01416-f007:**
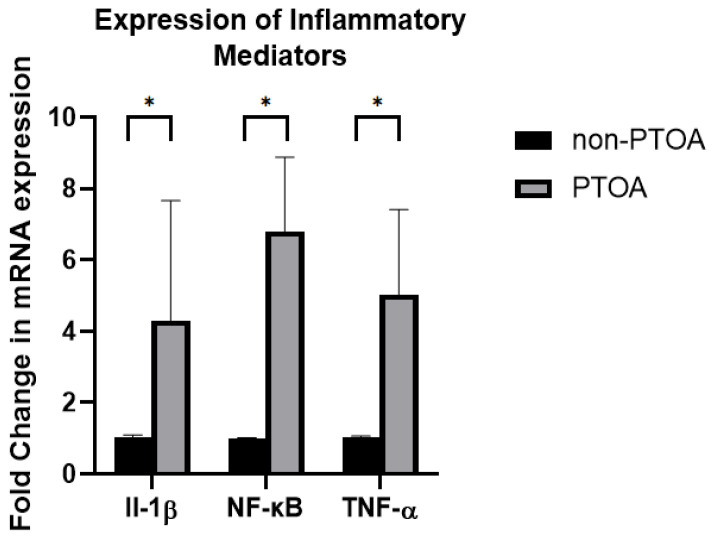
The mRNA expression of inflammatory mediators between non-PTOA and PTOA groups measured by the fold changes in IL-1β, NF-κB, and TNF-α. Error bars represent standard deviation. * Indicates significant difference between groups (*p* < 0.05).

**Figure 8 biomolecules-13-01416-f008:**
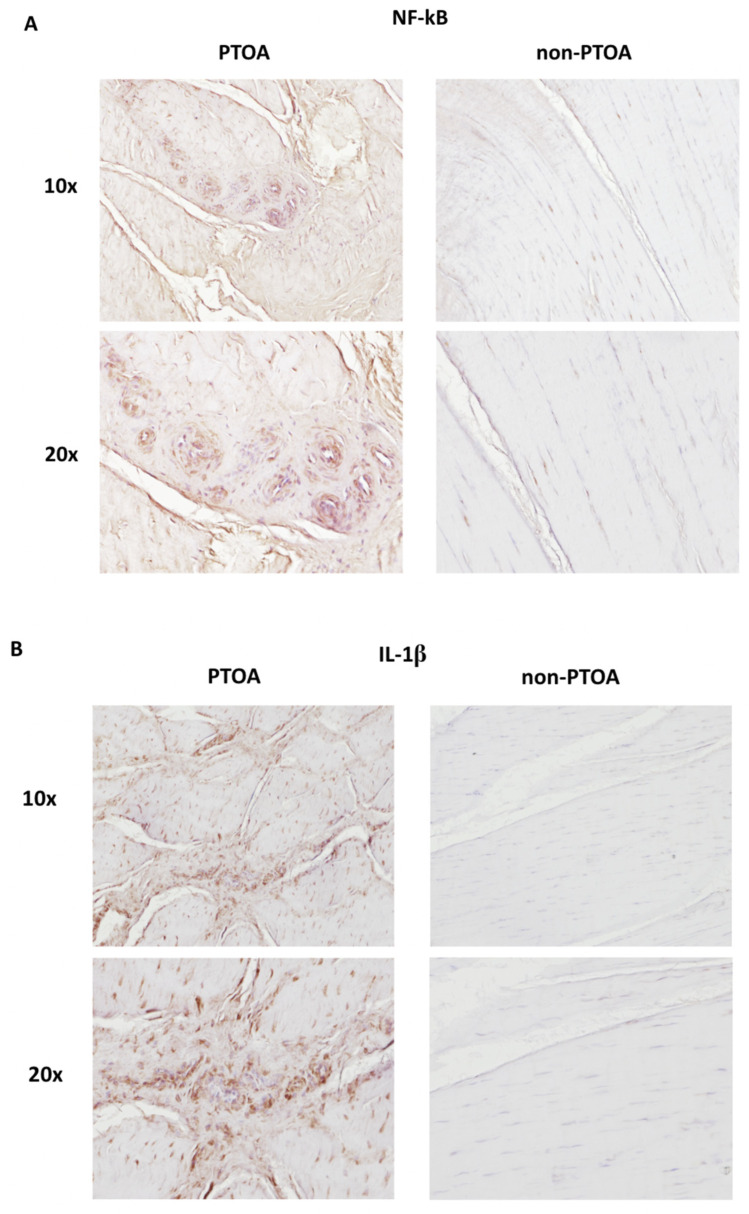
Immunohistochemistry staining of ACL tissue cells. (**A**) NF-κB staining for PTOA and non-PTOA groups. (**B**) IL-1β staining for PTOA and non-PTOA groups.

**Table 1 biomolecules-13-01416-t001:** Primers used for qRT-PCR evaluation of synovial membranes.

Target Gene	Primer Sequences	
IL-1β	Forward	5’-CTG CAA ACT CCA GGA CAA AGA-3
	Reverse	5’-GGG TGG CAT CAC AGA AAA-3’
NF-κB	Forward	5’-ACT TGC CAG ACA CAG ATG AC-3’
	Reverse	5’-GTC GGT GGG TCC ATT GAA A-3’
TNF-α	Forward	5’-CCT ACT GCA CTT CGA GGT TATC-3’
	Reverse	5’-ACG GGC TTA TCT GAG GTT TG-3’
18S	Forward	5’-GTAACCCGTTGAACCCCATT-3’
	Reverse	5’-CCATCCAATCGGTAGTAGCG-3’

## Data Availability

No new data were created or analyzed in this study. Data sharing is not applicable to this article.

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
