# Peer review of "The Effects of an Osteoarthritic Joint Environment on ACL Damage and Degeneration: A Yucatan Miniature Pig Model"

_biomolecules, 2023, doi:10.3390/biom13091416_

Round 1

Reviewer 1 Report

The authors reviewed 14 ACLs from 13 Yucatan minipigs, 7 of which had undergone modified intra-articular drilling (mIAD) procedure, which induces PTOA through inflammatory mediators. histological scoring systems. IL-1β, NF-κB, and TNF-α mRNA expression in the synovium was measured using qRT-PCR. Gait was analyzed using a pressure-sensing system. The authors should provide better images of the the cartilage and subchondral bone stainings. The use of Safranin O/Fast Green Stain would be appropriate. Also the authors should show high magnification images of the cartilage and subchondral bone. Immunohistochemistry should also be performed. As it is now it is just an observational description.

Author Response

Point 1: The authors should provide better images of the the cartilage and subchondral bone stainings. The use of Safranin O/Fast Green Stain would be appropriate. Also the authors should show high magnification images of the cartilage and subchondral bone

Point 1 Response: We have now utilized Safranin O/Fast Green stain to observe the changes in cartilage. We have provided  a detailed cartilage histological preparation methodology as well as images at varying and high magnifications. Additionally, we scored the cartilage using the OARSI system between PTOA and non-PTOA groups.

Point 2: Immunohistochemistry should also be performed.

Point 2 Response: We have now done immunohistochemistry on the ACL tissue that we analyzed in this study. We provided detailed methods describing the immunohistochemistry staining kit we used. Additionally, we have provided high magnification images of ACL tissue for NF-Kb staining and Il-1b staining, the results of which support our hypotheses and conclusions. 

Reviewer 2 Report

The goal of this study was to determine the effect of knee injury on the health of anterior cruciate ligament (ACL). For this purpose, the authors created two 15 mm holes/tunnels in the condyle and tibia plate. Then, they collected tissues at week-15 post surgeries. Gait abnormalities were assessed every 4 weeks and at week 15-post surgery. Then, the authors stated that despite the absence of intraoperative damage to soft tissues and no detectable gait disturbance, evaluation of cartilage integrity at 15 weeks following bone tunneling revealed dramatically increased cartilage degeneration and synovial inflammatory cells and cytokines compared to a sham surgery group. Histology measurement showed significant changes in the morphology of ACL at week-15-post surgery. 

Though the purpose of the study is of a great importance in the field of osteoarthritis, we have noticed several deficiencies with the design of the experiments that need to be addressed before the manuscript can be accepted for publication in Biomolecules Journal. 

Major comments

-          The authors have chosen week-15-post surgery to analyze different parameters, but one time point is for sure not enough to understand or explain the phenotype observed. In addition, there is no explanation for the rational for choosing this time point. Earlier time points are needed to determine if the damage in ACL was due to the first the drill or it is a consequence of cartilage damage post-surgery.

-          Cartilage damage and cytokine expression changes post-surgery need to be evaluated during the development of osteoarthritis (OA) at different time points to connect cause-consequence.

-          Without a time-course studies of the different changes at the injured knee joint, it is not possible to draw any conclusion.

-          The authors have used H&E and alcian blue to analyze the changes in ACL, while immunostaining using antibodies for different markers of inflammatory cells will be better to assess the inflammatory response of ACL to knee injury, and a biochemical analysis for ECM macromolecules can better assess ACL damage caused by knee injury. Please refer to Kharaz et al., (J. Anat., 2018, 232, pp. 943-955).

Minor comments

-          There was no mention of the healing of the hole/tunnel that was created at knee to cause the phenotype analyzed in this study. The authors need to show the position of the initial injury and its location in reference to cartilage damage position, you may circle the position of the initial injury in Fig. 1. Macroscopic changes.

-          Histology measurements need to be explained in detail, in terms of number of animals used and number of histology sections analyzed per sample/knee, …

Moderate editing of English language is required

Author Response

Point 1: The authors have chosen week-15-post surgery to analyze different parameters, but one time point is for sure not enough to understand or explain the phenotype observed. In addition, there is no explanation for the rational for choosing this time point. Earlier time points are needed to determine if the damage in ACL was due to the first the drill or it is a consequence of cartilage damage post-surgery. Cartilage damage and cytokine expression changes post-surgery need to be evaluated during the development of osteoarthritis (OA) at different time points to connect cause-consequence. Without a time-course studies of the different changes at the injured knee joint, it is not possible to draw any conclusion.

Point 1 Response: The study used a large preclinical animal model and it was expensive. We didn't have the budget to perform an early time point study at this time. However, other labs have demonstrated that loss of cartilage glycosaminoglycans and increased cartilage cell cloning scores were observed at 1-week and 4-week post-ACLR in the mini-pig ACLR model, and these changes were correlated with the up-regulation of proteases, such as MMP-1, -3, -13, ADAMTS4, ADAMTS5, IL-1β, IL-4, IL-6, and TNF-α . We have cited these studies in our revised manuscript as well as our recent publication (Sun et al., ) rationalizing the 15-week time point. Our purpose is to explore the effects of an osteoarthritic joint environment on ACL damage and degeneration at 15 weeks after surgery. We believe that the trauma-induced inflammation resulted in the changes of both cartilage and ACL at the same time. We appreciate the reviewer's comments and we will pay attention to the early changes of ACL and cartilage in a future study. However, we couldn't provide the earlier time points in ACL analysis for this study.

Point 2: The authors have used H&E and alcian blue to analyze the changes in ACL, while immunostaining using antibodies for different markers of inflammatory cells will be better to assess the inflammatory response of ACL to knee injury, and a biochemical analysis for ECM macromolecules can better assess ACL damage caused by knee injury. 

Point 2 Response: We appreciate the reviewers suggestions and have addressed them by performing immunohistochemistry. We have described in detail the staining kit and procedure. Additionally, we have provided high magnification images of the ACL tissue with immunostaining for NF-kB and Il-1B, the results of which support our hypotheses and conclusions. 

Point 3: There was no mention of the healing of the hole/tunnel that was created at knee to cause the phenotype analyzed in this study. The authors need to show the position of the initial injury and its location in reference to cartilage damage position, you may circle the position of the initial injury in Fig. 1. Macroscopic changes.

Point 3 Response: We have provided an image of the tibial and femoral condyles, indicating where the drill sites are located. 

Point 4: Histology measurements need to be explained in detail, in terms of number of animals used and number of histology sections analyzed per sample/knee, …

Point 4 Response: We appreciate the reviewers comments and have elaborated and specified our histological measurements. 

Round 2

Reviewer 2 Report

The authors have addressed some of the suggestions and comments, but I have noticed, in addition of some typo-mistakes, few disturbing issues in the present format of the manuscript. It is obvious that the authors have used the same animals used in their recently published manuscript (Sun et al., 2023) which is acceptable. However, presenting data in this manuscript that have already been published in their previous manuscript, is not acceptable, such as in the following figures. The authors will need to remove these data from this manuscript before it can be accepted for publication in Biomolecules Journal:

§  Figure 2 in this manuscript showed same data as figure 1 in Sun et al., (2023)

§  Figure 3 in this manuscript showed same data as in figure 2 from Sun et al., (2023)

§  Figure 9 in this manuscript showed data about some of the gait indicators that have been already published in their Sun et al (2023). I have noticed few differences in the shape of the graphs, but the authors did not provide any explanation…. Can the authors explain why they wanted to show graphs about the gait even though they have already published detailed data about different gait parameters from these same animals?

The manuscript needs extensive revision for language and typo-mistakes

Author Response

We greatly appreciate the time that you have taken to review this manuscript and thank you for providing detailed comments. Please find below detailed responses to address the reviewer's comments and corresponding revisions which have been made in the manuscript. 

Comment 1: Figure 2 in this manuscript showed same data as figure 1 in Sun et al., (2023)

Response 1: We acknowledge that the graphs between our current manuscript and the Sun et al. paper are similarly constructed and depict similar data. The pigs from our current paper comprise a subset of a larger group. However, the pigs shown between each paper vary. Sun et al. depict data from pigs labeled 8389 and 9126 where as in our current paper we show data from pigs 3020 and 3214. Additionally, the statistical analysis for the Sun et al. compares surgical to contralateral to sham legs whereas in our current study we are categorizing by PTOA vs non-PTOA animals. 

Comment 2:  Figure 3 in this manuscript showed same data as in figure 2 from Sun et al., (2023)

Response 2: Similarly to the first comment, we acknowledge the similarities in data presentation between the figures in each paper. This is a byproduct of the methods our lab uses to collect and depict data. However, the microscopic cartilage (Safranin O-fast green) staining in our current manuscript shows different pigs than the ones shown in the Sun et al. Additionally, the statistical analysis for our current paper compares PTOA and non-PTOA groups whereas Sun et al. compares surgical, contralateral, and sham legs. 

For both Figure 2 and Figure 3 of this manuscript, while we could remove them from the paper entirely, we believe that they are valuable supporting data that provide context our hypotheses and conclusions.

Comment 3: Figure 9 in this manuscript showed data about some of the gait indicators that have been already published in their Sun et al (2023). I have noticed few differences in the shape of the graphs, but the authors did not provide any explanation…. Can the authors explain why they wanted to show graphs about the gait even though they have already published detailed data about different gait parameters from these same animals?

Response 3: We agree that it is redundant to present such similar gait data to our previously published work. This is especially the case, as the fundamental aim of this paper was to examine the microscopic and inflammatory environments in osteoarthritis that can contribute to ACL degeneration. Accordingly, we have removed the gait data from this manuscript. 

Moderate Editing of English language required. 

Thank you for pointing this out. We have edited the manuscript accordingly, fixing typos and various grammatical errors.